# Thermal Sunyaev–Zel'dovich Effect in the IGM due to Primordial Magnetic Fields

**Teppei Minoda [1],[*],[†]** , **Kenji Hasegawa [1],[†]**, **Hiroyuki Tashiro [1],[†]**, **Kiyotomo Ichiki [1,2],[†]** and **Naoshi Sugiyama [1,2,3],[†]**

[1] Department of Physics and Astrophysics, Nagoya University, Nagoya 464-8602, Japan; hasegawa.kenji@a.mbox.nagoya-u.ac.jp (K.H.); hiroyuki.tashiro@nagoya-u.jp (H.T.); ichiki@a.phys.nagoya-u.ac.jp (K.I.); naoshi@nagoya-u.jp (N.S.)

[2] Kobayashi-Maskawa Institute for the Origin of Particles and the Universe, Nagoya University, Chikusa-ku, Nagoya 464-8602, Japan

[3] Kavli Institute for the Physics and Mathematics of the Universe (Kavli IPMU), The University of Tokyo, Chiba 277-8582, Japan

[*] Correspondence: minoda.teppei@d.mbox.nagoya-u.ac.jp

[†] These authors contributed equally to this work.

**Abstract:** In the present universe, magnetic fields exist with various strengths and on various scales. One possible origin of these cosmic magnetic fields is the primordial magnetic fields (PMFs) generated in the early universe. PMFs are considered to contribute to matter density evolution via Lorentz force and the thermal history of intergalactic medium (IGM) gas due to ambipolar diffusion. Therefore, information about PMFs should be included in the temperature anisotropy of the Cosmic Microwave Background through the thermal Sunyaev–Zel'dovich (tSZ) effect in IGM. In this article, given an initial power spectrum of PMFs, we show the spatial fluctuation of mass density and temperature of the IGM and tSZ angular power spectrum created by the PMFs. Finally, we find that the tSZ angular power spectrum induced by PMFs becomes significant on small scales, even with PMFs below the observational upper limit. Therefore, we conclude that the measurement of tSZ anisotropy on small scales will provide the most stringent constraint on PMFs.

**Keywords:** magnetic fields; large-scale structure; magnetic turbulence

## 1. Introduction

According to many astronomical observations, we can find the magnetic fields in the universe on a very wide range of spatial scales. The magnetic fields associate with different types of astrophysical objects, such as planets [1], ordinary stars [2], compact objects, star clusters, galaxies, clusters of galaxies [3], and so on. Additionally, some papers have suggested the existence of magnetic fields in the intergalactic region, also known as the cosmic voids, based on the blazars' $\gamma$-ray observations [4–7]. The origin of these cosmic magnetic fields is an open question, especially on large scales. Many scenarios are considered to explain the generation of such magnetic fields, and roughly speaking, such scenarios are divided into two groups, primordial origin and astrophysical origin. The first one indicates the mechanisms taking place in the primordial universe, and this scenario predicts tiny magnetic fields on cosmological scales. These weak seed fields are called "primordial magnetic fields" (PMFs). The latter entails that the cosmic magnetic fields were generated during or after large scale structure formation via mechanisms such as Biermann battery or Weibel instability.

The purpose of this study is to investigate how the PMFs affect structure formation after the recombination epoch, especially during the so-called Dark Ages. In addition, we estimate the Cosmic

Microwave Background (CMB) temperature anisotropy from thermal Sunyaev–Zel'dovich (tSZ) effect, in order to constrain the strength of PMFs with the cosmological observable. The calculation method described in this paper is based on our previous work [8].

## 2. Materials and Methods

### 2.1. Formalism of the PMFs

We treat the PMFs as the statistically-isotropic and homogeneous fields, so we can determine the characteristics of such fields from the power spectrum defined as:

$$\langle B_i^*(\mathbf{k}) B_j(\mathbf{k}') \rangle = \frac{(2\pi)^3}{2} \delta_D(\mathbf{k} - \mathbf{k}')(\delta_{ij} - \hat{k}_i \hat{k}_j) P_B(k) . \tag{1}$$

In this work, we assume $P_B(k)$ as a single power-law function of $k$ with the spectral index $n_B$ as:

$$P_B(k) = \frac{n_B + 3}{2} \frac{(2\pi)^2 B_n^2}{k_n^{n_B+3}} k^{n_B}, \tag{2}$$

where $B_n$ is the field strength at the normalized scale $k_n \equiv 2\pi \ \mathrm{Mpc}^{-1}$.

By taking the top-hat window function in Fourier space, we can find the relation between $B_n$ and $B_\lambda$, which is the amplitude of the PMF smoothed on any other spatial scale $\lambda$ as:

$$B_\lambda^2 = \int_0^{k_\lambda} P_B(k) \frac{d^3k}{(2\pi)^3} = B_n^2 \left( \frac{k_\lambda}{k_n} \right)^{n_B+3} , \tag{3}$$

with $k_\lambda = 2\pi/\lambda$.

According to some previous works [9,10], we assume the cut-off scale of the PMFs caused by the damping of Alfvén waves. This cut-off wave number, $k_c$, is given by:

$$k_c^{-2} = \frac{B_{\lambda_c}^2(t_{\mathrm{rec}})}{4\pi \rho_\gamma(t_{\mathrm{rec}})} \int_0^{t_{\mathrm{rec}}} \frac{l_\gamma(t')}{a^2(t')} dt', \tag{4}$$

where $t_{\mathrm{rec}}$, $\rho_\gamma$, $l_\gamma$, and $a$ are the recombination time, the energy density of CMB photon, the mean-free path of the CMB photon, and the cosmic scale factor, respectively. Furthermore, $\lambda_c = 2\pi/k_c$ shows the cut-off wavelength of PMFs.

For $k \leq k_c$, the power spectrum of PMFs is given by Equation (2), and PMFs are completely damped for $k > k_c$. Thus, we put the cut-off scale only in the ultraviolet regime. We discuss the effect of the large-scale fluctuation of PMFs in Section 4. Furthermore, the time evolution of PMFs is assumed as $\mathbf{B}(t, \mathbf{x}) = \mathbf{B}(t_{\mathrm{now}}, \mathbf{x})/a^2(t)$, and we neglect the helicity of PMFs for simplicity.

When we fix the values of $B_n$ and $n_B$, the amplitude of the power spectrum of the PMFs can be obtained by Equation (2), and the cut-off scale is calculated through Equations (3) and (4). Therefore, in this model, the parameters $B_n$ and $n_B$ determine the statistical property of PMFs. We study the tSZ effect induced by PMFs in the case of $B_n = 0.5$ nG and $n_B = -1$, which are not excluded from the Planck constraint on PMFs [11]. In this work, we perform the numerical calculation of matter density evolution and the thermal evolution of the intergalactic medium (IGM) gas including the PMFs. We thus obtain the CMB angular power spectrum with the effect of PMFs via tSZ taken into account. In the next three subsections, we explain the basic equations we used.

### 2.2. Matter Density Evolution with PMFs

When the density contrast $\delta$ is smaller than unity, the time evolutional equations of overdensities can be written in terms of the background baryon density $\rho_b$ and the cold dark matter density $\rho_c$, as the following [12]:

$$\frac{\partial^2 \delta_c}{\partial t^2} + 2H(t)\frac{\partial \delta_c}{\partial t} - 4\pi G(\rho_c \delta_c + \rho_b \delta_b) = 0, \tag{5}$$

$$\frac{\partial^2 \delta_b}{\partial t^2} + 2H(t)\frac{\partial \delta_b}{\partial t} - 4\pi G(\rho_c \delta_c + \rho_b \delta_b) = S(t), \tag{6}$$

where $H(t)$ is the Hubble parameter and subscripts "b" and "c" represent the values for the baryon and the cold dark matter, respectively. $S(t)$ is the source term for the baryon density fluctuation $\delta_b$ due to Lorentz force, given by:

$$S(t,\mathbf{x}) = \frac{\nabla \cdot (\nabla \times \mathbf{B}(t,\mathbf{x})) \times \mathbf{B}(t,\mathbf{x})}{4\pi \rho_b(t) a^2(t)}, \tag{7}$$

where $\mathbf{B}(t,\mathbf{x})$ denotes the PMFs in a comoving three-dimensional space $\mathbf{x}$ and time $t$ and $\nabla$ is also taken in comoving coordinates. Assuming a matter dominated era and $\delta_b = 0$ during the recombination epoch, we obtain the solution of Equation (6) as:

$$\delta_b = \frac{2S(t)}{15H^2(t)}\left[\left\{3\left(\frac{a}{a_{\text{rec}}}\right) + 2\left(\frac{a}{a_{\text{rec}}}\right)^{-\frac{3}{2}} - 15\ln\left(\frac{a}{a_{\text{rec}}}\right)\right\}\frac{\Omega_b}{\Omega_m} + 15\ln\left(\frac{a}{a_{\text{rec}}}\right) + 30\left(1 - \frac{\Omega_b}{\Omega_m}\right)\left(\frac{a}{a_{\text{rec}}}\right)^{-\frac{1}{2}} - \left(30 - 25\frac{\Omega_b}{\Omega_m}\right)\right], \tag{8}$$

where $\Omega_m$ is the density parameter for total matter and $\Omega_b$ is the one for baryonic matter. Furthermore, $a_{\text{rec}} \equiv a(t_{\text{rec}})$ is the scale factor at the recombination time $t_{\text{rec}}$. In this work, we set the recombination epoch at $z_{\text{rec}} \equiv 1089$. Note that Equations (5) and (6) are valid only when $\delta_b \ll 1$ and baryonic pressure is negligible. The validity of these assumptions will be discussed in the final section.

*2.3. Thermal Evolution with PMFs*

According to [12], PMFs not only induce the density fluctuation of IGM, but also affect the thermal history of IGM via an energy dissipation mechanism, so-called ambipolar diffusion. The equation of IGM gas temperature $T_{\text{gas}}$ evolution with heating due to the ambipolar diffusion is given by [13]:

$$\begin{aligned}
\frac{dT_{\text{gas}}}{dt} = & - 2H(t)T_{\text{gas}} + \frac{\dot{\delta}_b}{1+\delta_b}T_{\text{gas}} \\
& + \frac{x_e}{1+x_e}\frac{8\rho_\gamma \sigma_T}{3m_e c}(T_\gamma - T_{\text{gas}}) + \frac{\Gamma(t)}{1.5k_B n_b} \\
& - \frac{x_e n_b}{1.5k_B}[\Theta x_e + \Psi(1-x_e) + \eta x_e + \zeta(1-x_e)],
\end{aligned} \tag{9}$$

where $x_e$, $m_e$, $\sigma_T$, $T_\gamma$, $k_B$, and $n_b$ are the ionization fraction of hydrogen atoms, the rest mass of an electron, the Thomson cross-section, the CMB temperature, the Boltzmann constant, and the number density of the baryon gas, respectively. The first term in the right-hand side of Equation (9) is adiabatic cooling due to the cosmic expansion, and the second is the effect of adiabatic compression (or expansion) from local density fluctuation. The third one is the energy transfer between CMB photons and thermal electrons, the so-called Compton heating (or cooling). The forth term with $\Gamma(t)$ represents the extra heating source for the IGM, which is injected by the PMF dissipation through ambipolar diffusion. We adopt the expression for $\Gamma(t)$, which means the heating energy per unit time per unit volume, as [12]:

$$\Gamma(t,\mathbf{x}) = \frac{|(\nabla \times \mathbf{B}(t,\mathbf{x})) \times \mathbf{B}(t,\mathbf{x})|^2}{16\pi^2 \xi \rho_b^2(t)}\frac{(1-x_e)}{x_e}, \tag{10}$$

where $\zeta$ is the drag coefficient for H and H$^+$, and here, the value of $\zeta = 3.5 \times 10^{13}$ cm$^3$g$^{-1}$s$^{-1}$ [14]. The last term of Equation (9) with the coefficients $\Theta$, $\Psi$, $\eta$, and $\zeta$ is the radiative cooling effects for hydrogen atoms. We use the definitions and values from [13].

In order to solve Equation (9), we have to follow the evolution of the ionization fraction, as well,

$$\frac{dx_e}{dt} = \frac{1 + K_\alpha \Lambda n_b (1 - x_e)}{1 + K_\alpha (\Lambda + \beta_e) n_b (1 - x_e)} \times \left[ -\alpha_e n_b x_e^2 + \beta_e (1 - x_e) \exp\left( -\frac{3E_{\text{ion}}}{4 k_B T_\gamma} \right) \right] + \gamma_e n_b x_e , \tag{11}$$

where $K_\alpha$, $\Lambda$, $\alpha_e$, $\beta_e$, and $\gamma_e$ are the parameters for the ionization and recombination processes, and they are given as the functions of $T_{\text{gas}}$ in [15–17]. Therefore, we calculate Equations (9) and (11) simultaneously, and we take into account the fluctuations of the IGM density $\delta_b$, given by Equation (8). For simplicity, we neglect the presence of helium, heavier elements and, any astronomical objects.

*2.4. CMB Angular Power Spectrum*

As explained above, the tangled PMFs could generate spatial fluctuations of the IGM number density $n_b$, ionization fraction $x_e$, and temperature $T_{\text{gas}}$. They can induce the CMB temperature anisotropy via the inverse Compton scattering, which is called the tSZ effect. In this subsection, we derive the angular power spectrum of the CMB temperature caused by the tSZ effect.

The strength of tSZ effect is represented by the Compton $y$-parameter integrated along a line-of-sight $\hat{n}$ [18],

$$y(\hat{n}) \equiv \frac{k_B \sigma_T}{m_e c^2} \int dz \frac{c w(\hat{n}, z)}{H(1 + z)}, \tag{12}$$

In Equation (12), $w(\hat{n}, z)$ is the convolutional function of $n_b$, $x_i$, and $T_{\text{gas}}$ as,

$$w(\hat{n}, z) = \left[ x_e n_b (T_{\text{gas}} - T_\gamma) \right]_{\hat{n}, z} . \tag{13}$$

The CMB temperature anisotropies from the tSZ effect are related to the Compton $y$-parameter as:

$$\frac{\Delta T}{T}(\hat{n}) = g_\nu y(\hat{n}), \tag{14}$$

where $g_\nu$ is a function of the frequency where the CMB is observed, and is given by $g_\nu = -4 + x / \tanh(x/2)$ with $x \equiv h_{\text{Pl}} \nu / k_B T$. Therefore, we obtain the CMB angular power spectrum as:

$$\mathcal{D}_\ell = \frac{\ell(\ell + 1)}{2\pi} \left( \frac{g_\nu k_B \sigma_T}{m_e c^2} \right)^2 \int d\chi \frac{P_w(\chi, \ell/\chi)}{\chi^2}, \tag{15}$$

where $\ell$ is the multipole and $P_w(\chi, k)$ is the power spectrum of the Compton $y$-parameter for two points, with a separation corresponding to wavenumber $k$ at comoving distance $\chi$. We can obtain $P_w(\chi, k)$ from $w$ via Equation (13). We have adopted Limber's approximation when we derive Equation (15) because we are interested in only large $\ell$ modes, where the tSZ signal dominates the primary CMB anisotropy.

**3. Results**

We performed our simulations for $B_n = 0.5$ nG and $n_B = -1$. The left panel of Figure 1 shows the two-dimensional structure of the x-component of the Lorentz force $(\nabla \times \mathbf{B}) \times \mathbf{B}$, which appears in Equations (7) and (10). The middle and right panels show the baryon gas temperature and the baryon number density at $z = 10$, respectively. The size of each panel is $2 \times 2$ comoving Mpc$^2$. You can see that the gas temperature rises to ~20,000 K due to the ambipolar diffusion in several places. In such regions, the baryon gas density is decreased due to the Lorentz force. Thus, $T_{\text{gas}}$ and $n_b$ have

a negative correlation. The reason for this anti-correlation is seen in Equation (10): the heating rate from the ambipolar diffusion is proportional to $\rho_b^{-2}$. We also have found that the density fluctuation of the baryon gas exceeds unity soon after the recombination epoch. Therefore, in order to avoid the negative mass density, we put the lowest bound on density fluctuation of baryon gas as $\delta_b = -0.9$.

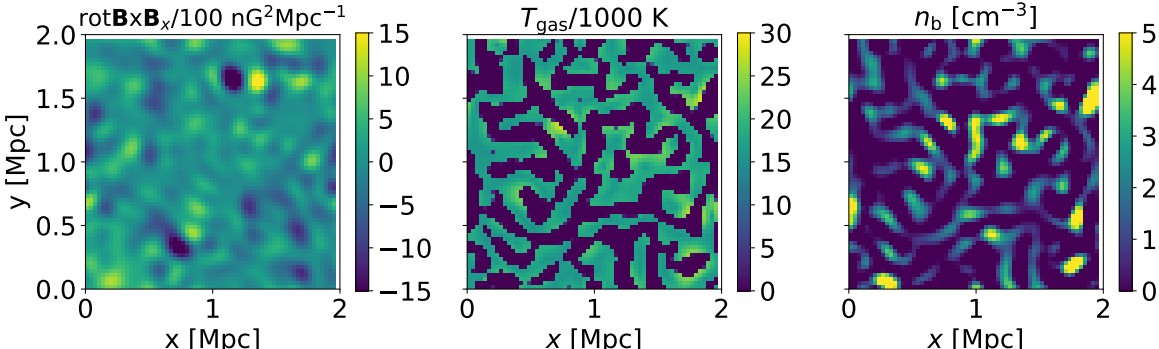

**Figure 1.** Illustration of $(\nabla \times \mathbf{B}) \times \mathbf{B}_x$ (left), the gas temperature (middle), and the gas number density (right) when $z = 10$. You can see the spatial anti-correlation between $T_{gas}$ and $n_b$.

In Figure 2, we plot the CMB temperature angular power spectrum $\mathcal{D}_\ell$ caused by the tSZ effect in the IGM due to PMFs. The tSZ angular power spectrum has a sharp peak around the multipole $\ell \sim 10^6$. This angular scale corresponds to the PMF cut-off scale, which is of the order 100 comoving kpc. The current observation can reveal the CMB angular power spectrum only for $\ell \lesssim 10^4$, but the tSZ angular power spectrum from PMFs dominates the primary one on smaller scales. This is because the primary component of CMB anisotropy has experienced Silk damping, but the PMFs can continue to create fluctuations of physical variables of baryon gas after the recombination epoch. Therefore, the information about the PMF cut-off scale is included in the small-scale CMB anisotropy, due to the tSZ effect in the IGM. The future ground-based observations, such as "CMB-S4", would give us much information on such a small-scale CMB anisotropy.

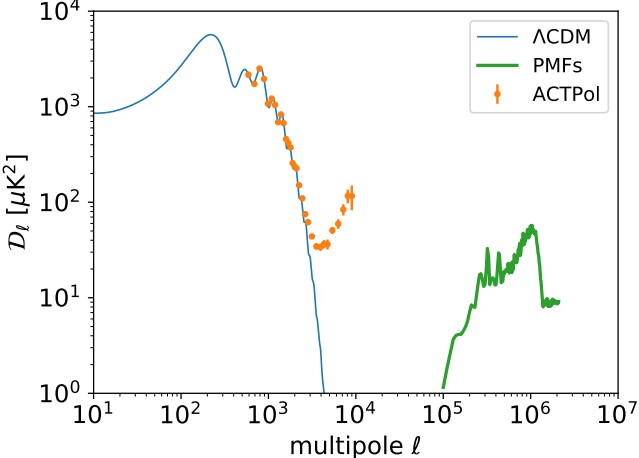

**Figure 2.** CMB temperature power spectrum for the multipole $10 < \ell < 10^7$. We plot the resultant angular power spectrum caused by the tSZ effect from the PMFs with the thick green line. The result is obtained by integrating Equation (15) from $z = 1000$ to $z = 10$, and the parameters for the PMFs are fixed to $B_n = 0.5$ nG and $n_B = -1$. The primary component of the CMB angular power spectrum predicted from the ΛCDM cosmology and the observational data with the Atacama Cosmology Telescope (two-season ACTPol) [19] are also shown with the blue solid line and the orange dots with error bars, respectively.

## 4. Discussions and Conclusions

In this section, we discuss the validity of the setup and assumptions in this study. In this work, we have fixed the box size as (2 comoving Mpc)$^3$ and the parameters of the PMF model as $B_n = 0.5$ nG and $n_B = -1$. Although we do not show it in this paper, we have checked the dependence of our results on these parameters. When we enlarged the simulation box size and calculated the resultant tSZ signal, we obtained the same amplitude and shape of tSZ signal as those in the fiducial box size. Next, we followed the IGM evolution with various spectral indices of the PMFs in the range $-1 \leq n_B \leq 2$. As a result, we have also obtained a tSZ signal with a similar shape, but we have found that the signal amplitude is determined by the cut-off length of the PMFs, which depends on the values of $B_n$ and $n_B$.

Next, we discuss the treatment of the matter density evolution. As seen in Section 2.2, we have assumed two important conditions to derive Equations (5) and (6). First of all, we neglect the thermal pressure of the baryon gas, and this condition is satisfied on scales larger than the Jeans scale of the fluid. We have confirmed that the minimum grid length of our calculation is always greater than the Jeans length. Next, we found that the local density perturbation is greater than unity soon after recombination, even for the sub-nano Gauss PMF model. However, the contribution to the tSZ power spectrum is largely created by the low density region, because of the anti-correlation between the electron density and temperature. Therefore, this overestimation of the local matter density is not a severe problem for our results. In order to confirm this point, we will use the cosmological MHD simulation to estimate such a non-linear effect on the tSZ signal instead of linear perturbation theory in our future work. Such a simulation is also thought to reveal highly non-linear MHD dissipation mechanisms, in addition to the ambipolar diffusion, as recently discussed for the recombination epoch [20].

In this work, we have investigated the tSZ signal caused by the PMFs in the so-called Dark Ages. By assuming the power spectrum of the PMFs as $P_B(k) \propto B_n^2 k^{n_B}$, with $B_n = 0.5$ nG and $n_B = -1$, we consistently solve the evolutionary equations for the IGM gas density and temperature. At last, we estimate the tSZ angular power spectrum caused by the fluctuations of IGM gas with sub-nano Gauss PMFs. We found that the PMFs create an anti-correlation between the gas temperature and density. We have shown that tSZ signal has a sharp peak at $\ell \sim 10^6$, which corresponds to the cut-off scale of PMFs due to the radiative viscosity in the early universe. Therefore we can conclude that the observation of the small scale CMB temperature anisotropy could give us nature of the cosmic magnetic fields.

**Author Contributions:** Conceptualization, H.T., K.I.; methodology, T.M., K.H., H.T., K.I.; validation, K.H., K.I.; formal analysis, T.M., K.I.; investigation, T.M., K.H.; writing, original draft preparation, T.M.; writing, review and editing, H.T., K.I., N.S.; visualization, T.M.; supervision, N.S.; project administration, T.M., H.T., N.S.; funding acquisition, H.T., K.I., N.S.

**Funding:** This research is supported by the KAKEN Grant-in-Aid for Scientific Research, (B) No. 25287057 (K.I. and N.S.), (A) No. 17H01110 (N.S.), and 16H01543 (K.I.) and for Young Scientists (B) No. 15K17646 (H.T.).

**Acknowledgments:** We thank the local and scientific organizing committee for the international conference, "THE POWER OF FARADAY TOMOGRAPHY—TOWARDS 3D MAPPING OF COSMIC MAGNETIC FIELDS—". We also appreciate the anonymous referees and editor, and we give special thanks to Patel Teerthal for careful reading of our paper.

**Conflicts of Interest:** The authors declare no conflict of interest. The founding sponsors had no role in the design of the study; in the collection, analyses, or interpretation of data; in the writing of the manuscript; nor in the decision to publish the results.

## Abbreviations

The following abbreviations are used in this manuscript:

| | |
|---|---|
| MDPI | Multidisciplinary Digital Publishing Institute |
| CMB | Cosmic Microwave Background |
| PMF | primordial magnetic field |
| IGM | intergalactic medium |
| tSZ | thermal Sunyaev–Zel'dovich effect |

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
