# Peer review of "Thermal Sunyaev–Zel’dovich Effect in the IGM due to Primordial Magnetic Fields"

_galaxies, doi:10.3390/galaxies6040143_

Round 1
Reviewer 1 Report
This paper describes how primordial magnetic fields affect matter density and temperature distribution of the intergalactic medium on small scales and how constraints on primordial magnetic field strength can be derived from the measurement of the Cosmic Microwave Background temperature anisotropy through the thermal Sunyaev-Zel'dovitch effect. The work is interesting and worth for publication, especially in view of future observations.
I recommend this paper for publication after the following minor revisions:
- please, state after each formula/equation the meaning of all the quantities involved (e.g., for Eq. 4 ργ and a(t) should be defined);
- please describe the physical meaning and importance of Bλ and of λ (see Eq. 3);
- in section 2.1 the authors consider a power spectrum defined over a wave-number range0
- please, explain the reason of the choice nB=-1 and comment on how the results would change for different strengths of the magnetic fields and for different slopes of the power spectrum;
- It would be useful if the authors could comment (in section 3) about future instrument/observation that can be used to measure the effects they describe;
- I recommend a revision from a native English speaker.
Author Response
Dear Referee,
Thank you very much for your message of 22nd November 2018. Together with a reply to referee’s comments, we are resubmitting our paper “Thermal Sunyaev-Zel'dovich effect in the IGM due to the primordial magnetic fields”.
We appreciate referee’s careful reading of our paper and helpful suggestions.
Our replies to referee’s comments are listed below. The modifications in the revised manuscript are highlighted in red. We hope that our paper is now suitable for publication in Galaxies.
Sincerely,
Teppei Minoda, Kenji Hasegawa, Hiroyuki Tashiro, Kiyotomo Ichiki, and Naoshi Sugiyama

Reviewer 2 Report
The paper is well-structured and reads very well. I only found minor editorial corrections as mentioned below.
One minor suggestion: it might be worth mentioning the recent work by Trivedi et al., 2018 (http://adsabs.harvard.edu/abs/2018MNRAS.481.3401T) which considered the evolution of magnetic fields across the recombination epoch. Real heating of the medium only occurs in the turbulent regime and as also pointed out there, this creates secondary velocity and density perturbations. The anisotropies in the temperature of the medium could be affected by these effects and hence the shape of the small scale power spectrum. I leave it to the authors to decide about adding a brief discussion.
- introduction: "The last one includes..." --> "The latter includes"
- line 103: "...soon after the recombination" --> "...soon after recombination"
Author Response
Dear Referee,
Thank you very much for your message of 12th November 2018. Together with a reply to referee’s comments, we are resubmitting our paper “Thermal Sunyaev-Zel'dovich effect in the IGM due to the primordial magnetic fields”.
We appreciate referee’s careful reading of our paper and helpful suggestions.
Our replies to referee’s comments are attached as a Word file named "replies_for_reviewer2.docx". The modifications in the revised manuscript are highlighted in red. We hope that our paper is now suitable for publication in Galaxies.
Sincerely,
Teppei Minoda, Kenji Hasegawa, Hiroyuki Tashiro, Kiyotomo Ichiki, and Naoshi Sugiyama
